# Pemafibrate Protects against Fatty Acid-Induced Nephropathy by Maintaining Renal Fatty Acid Metabolism

**DOI:** 10.3390/metabo11060372

**Published:** 2021-06-09

**Authors:** Daiki Aomura, Makoto Harada, Yosuke Yamada, Takero Nakajima, Koji Hashimoto, Naoki Tanaka, Yuji Kamijo

**Affiliations:** 1Department of Nephrology, Shinshu University School of Medicine, Matsumoto, Nagano 390-8621, Japan; aomura91@gmail.com (D.A.); tokomadaraha724@gmail.com (M.H.); yosukeyama@shinshu-u.ac.jp (Y.Y.); khashi@shinshu-u.ac.jp (K.H.); 2Department of Metabolic Regulation, Shinshu University School of Medicine, Matsumoto, Nagano 390-8621, Japan; nakat@shinshu-u.ac.jp (T.N.); naopi@shinshu-u.ac.jp (N.T.); 3International Relations Office, Shinshu University School of Medicine, Matsumoto, Nagano 390-8621, Japan

**Keywords:** pemafibrate, peroxisomal proliferator-activated receptor alpha, renal fatty acid metabolism, lipotoxicity, nephrology

## Abstract

As classical agonists for peroxisomal proliferator-activated receptor alpha (PPARα), fibrates activate renal fatty acid metabolism (FAM) and provide renoprotection. However, fibrate prescription is limited in patients with kidney disease, since impaired urinary excretion of the drug causes serious adverse effects. Pemafibrate (PEM), a novel selective PPARα modulator, is mainly excreted in bile, and, thus, may be safe and effective in kidney disease patients. It remains unclear, however, whether PEM actually exhibits renoprotective properties. We investigated this issue using mice with fatty acid overload nephropathy (FAON). PEM (0.5 mg/kg body weight/day) or a vehicle was administered for 20 days to 13-week-old wild-type male mice, which were simultaneously injected with free fatty acid (FFA)-binding bovine serum albumin from day 7 to day 20 to induce FAON. All mice were sacrificed on day 20 for assessment of the renoprotective effect of PEM against FAON. PEM significantly attenuated the histological findings of tubular injury caused by FAON, increased the renal expressions of mRNA and proteins related to FAM, and decreased renal FFA content and oxidative stress. Taken together, PEM exhibits renoprotective effects through the activation and maintenance of renal FAM and represents a promising drug for kidney disease.

## 1. Introduction

Peroxisomal proliferator-activated receptor α (PPARα) is a member of the steroid/nuclear receptor superfamily. PPARα is distributed in fatty acid-consuming organs such as the kidney and liver, and the activation of PPARα signaling increases the expression of its target genes, encoding enzymes involved in fatty acid metabolism (FAM) [1,2]. In the kidney, PPARα is mainly expressed in proximal tubular epithelial cells (PTECs) to maintain energy homeostasis and tubular reabsorptive function [3].

PPARα expression and FAM are markedly reduced in the injured kidney [4]. A renal FAM deficiency causes excess lipid accumulation in PTECs, further kidney injury, and the progression of renal dysfunction through lipotoxicity (LTx) [5]. Based on these facts, the activation of PPARα signaling and FAM in the kidney is presumed to suppress renal fibrosis and the progression of renal dysfunction in patients with chronic kidney disease (CKD) [6]. 

As classical PPARα agonists, fibrates reportedly suppress microalbuminuria in patients with diabetic nephropathy [7,8,9]. However, the drugs are primarily excreted in urine, and their serum concentration can quickly become elevated in patients with CKD, with severe accompanying adverse effects including renal toxicity and rhabdomyolysis [7]. It is therefore risky to prescribe fibrates in such patients, and no safe clinical methods to activate renal PPARα signaling have been established to date.

Pemafibrate (PEM) was first approved in Japan in 2017. This novel drug is pharmacologically defined as a selective PPARα modulator (SPPARMα), with a mode of PPARα activation different from those of classical fibrates. SPPARMα agents have a Y-shaped structure, formed by the introduction of benzoxazole and phenoxyalkyl sidechains, which differs remarkably from the linear structure of classic fibrates [10]. The unique SPPARMα structure enables strong and complete binding to the Y-shaped ligand-binding pocket of PPARα [11], producing marked increases in potency and higher subtype selectivity for PPARα [12]. In vivo experimental models have shown superior lipid-modifying activity with PEM as compared with other fibrates in terms of lowering triglyceride (TG) levels and raising high-density lipoprotein (HDL) levels [12].

PEM rarely causes adverse effects and is safe even in patients with CKD due to its biliary mode of excretion [13,14,15]. The incidence of adverse events by PEM in CKD patients was found to be similar to that by a placebo [16]. Although PEM is considered safe and efficient for kidney disease, it remains unclear whether it can actually exert renoprotective effects. To clarify this issue, we administered PEM to fatty acid overload nephropathy (FAON) model mice and assessed its protective properties in the kidney.

## 2. Results

### 2.1. PEM Increased the Renal Expressions of PPARα Target Genes

This study consisted of two independent experiments. In experiment one, PEM (0.5 mg/kg body weight/day) was administered for 14 days to eight-week-old wild-type (WT) male mice (PEM group) for comparisons with vehicle-administered mice (Veh group). Blood tests showed that PEM remarkably decreased serum TG levels and increased serum HDL levels without elevating serum alanine aminotransferase (ALT) or creatinine (Cr) levels (Appendix A). Furthermore, the renal expressions of mRNA and proteins related to FAM were significantly enhanced in the PEM group as compared with the Veh group (Figure 1). Especially, FAM in peroxisomes, i.e., L-peroxisomal bifunctional protein (PH) and peroxisomal 3-ketoacyl-CoA thiolase (PT), were markedly increased. These results indicated that PEM improved lipid metabolism and activated renal FAM without obvious adverse effects. 

#### 2.1.1. PEM Attenuated Tubular Injury and Urine Findings in FAON Model Mice

In experiment two, we assessed the effect of PEM on FAON model mice. PEM (0.5 mg/kg body weight/day) or a vehicle was administered to 13-week-old WT mice every day, and a daily intraperitoneal bolus injection of free fatty acid-binding bovine serum albumin (FFA-BSA) was started on day seven to induce FAON [17,18]. Mice were divided into three groups: vehicle-treated mice with sterile saline injection (controls; CON group), vehicle-treated mice with FAON (FAON group), and PEM-treated mice with FAON (PEM + FAON group). Since experiment one compared the vehicle and PEM groups and indicated no apparent differences apart from factors related to FAM (Appendix A), we used only the vehicle group with a sterile saline injection as the control group. All mice were sacrificed on day 20. 

Light microscopic analysis showed the findings of severe tubular injury consisting of tubular dilation and sloughing of PTECs in the FAON group, whereas these results were less evident in the PEM + FAON group (Figure 2A). Semi-quantitative scoring analysis also demonstrated the severity of tubular injury to be significantly improved in the PEM + FAON group (Figure 2B). Immunohistochemical investigation showed strong positive staining for osteopontin, a pathological marker of tubular injury, in PTECs in the FAON group, but only weak staining in the PEM + FAON group (Figure 2C). Electron microscopy revealed the swelling of mitochondria in PTECs in the FAON group, which was scarcely detected in the PEM + FAON group (Figure 2D). Although blood urea nitrogen (BUN) levels were comparable between the FAON and PEM + FAON group, they were significantly increased in both groups over controls (Figure 3A). Serum Cr levels were similar among the groups. The urine total protein/urine Cr ratio and the urine N-acetyl-β-D-glucosaminidase (NAG)/urine Cr ratio were greatly increased by FAON, with elevations of these urine markers significantly milder in the PEM + FAON group (Figure 3B,C).

#### 2.1.2. PEM Improved Renal FAM in FAON Model Mice

We next assessed the changes in FAM by PEM in FAON model mice. Serum and renal FFA levels were dramatically elevated in the FAON group and decreased in the PEM + FAON group (Figure 4A). Renal adenosine triphosphate (ATP) content was decreased in the FAON group and increased in the PEM + FAON group (Figure 4B). The renal expressions of mRNA and proteins related to FAM were also evaluated (Figure 4C,D). Renal FAM in peroxisomes, i.e., acyl-CoA oxidase (ACOX), PH, and PT, was dramatically decreased in the FAON group and improved in the PEM + FAON group. Renal FAM in mitochondria, i.e., carnitine palmitoyl-transferase 2 (CPT2), very long-chain acyl-CoA dehydrogenase (VLCAD), medium-chain acyl-CoA dehydrogenase (MCAD), and mitochondrial trifunctional protein α subunits (TPα), exhibited similar changes, although differences between the FAON and PEM + FAON group were not significant. These results suggested that the renoprotective effect of PEM was caused by the maintenance of renal FAM against FAON.

#### 2.1.3. PEM Reduced Oxidative Stress in the Kidneys of FAON Model Mice

Since tubular injury in FAON model mice was related to intracellular LTx-causing oxidative stress (OS) [5,17], we examined OS markers and the expressions of antioxidant agents in the kidney. Immunoblot analysis revealed that a marker of tissue OS, 4-hydroxynonenal (4-HNE), was significantly increased in the FAON group but was attenuated in the PEM + FAON group (Figure 5A). The renal expressions of mRNA and proteins related to antioxidant enzymes were examined next (Figure 5B,C). Although the renal expressions of superoxide dismutase 1 (SOD1) and catalase did not differ remarkably, those of superoxide dismutase 2 (SOD2) were significantly decreased in the FAON group and improved in the PEM + FAON group.

## 3. Discussion

The present study clearly demonstrated that PEM attenuated FFA-induced tubular injury by maintaining renal FAM and decreasing OS in the kidneys.

In patients with CKD, the expressions of PPARα target genes related to FAM in PTECs are decreased, with deficiencies in renal FAM associated with the progression of renal dysfunction [6]. The mechanism of how defects in FAM cause renal dysfunction is controversial. Several studies have reported that FFA-binding urine albumin plays a key role in this pathomechanism [5,19]; the damaged glomeruli in CKD pass a large amount of serum albumin and bound FFA (Alb-FFA) to the proximal tubular lumens, most of which is reabsorbed into PTECs. However, FAM capacity in PTECs is impaired in CKD patients, causing excessive FFA accumulation. Excess FFA leads to LTx, inducing the collapse of energy homeostasis, increased OS in PTECs, and, ultimately, a decline in renal function [5]. FAON model mice simulate the above mechanism by the injection of excessive Alb-FFA. Since PTECs in this model are overloaded with FFAs without glomerular injury, FAON mice are suitable for direct assessment of the relationship between defective FAM in PTECs and tubular injury [17]. 

The activation of PPARα signaling and FAM in the kidney by fibrates have been identified as renoprotective in animal experiments [18,20,21]. Furthermore, a systematic review of randomized clinical trials found that fibrates improved lipid profiles and reduced albuminuria progression, a major risk factor of renal death, in patients with diabetic nephropathy [7]. In patients with CKD, however, serum fibrate concentrations are easily elevated, since they are primarily excreted in urine, and various adverse effects frequently ensue [7]. Fibrates are also known to increase serum Cr levels. The precise mechanism of this elevation is unknown, and several studies do not consider it a loss of renal function [7,9,22]. However, the serum Cr elevation reportedly increases nephrology consultations and patient hospitalizations [23], and KDIGO guidelines recommend the very careful prescription of fibrates in patients with CKD [24]. It is a dilemma not to readily prescribe fibrates for CKD patients despite the drug’s renoprotective benefits. As a novel SPPARMα agent, PEM specifically activates PPARα with few adverse effects [13,14]. PEM has been considered safe even in patients with CKD because of its biliary excretion [15,25], and did not cause serum Cr elevation in a phase three trial, different from other fibrates [25]. In the present study, PEM attenuated tubular injury in FAON model mice, which suggested it to be safe and renoprotective in patients with CKD.

The severity of tubular injury is closely associated with renal prognosis [26,27,28]. In experiment two, the pathological findings of tubular injury were dramatically attenuated by PEM, strongly indicating a renoprotective effect. Although the levels of BUN and serum Cr were unaffected by PEM, those molecular markers are reportedly insensitive to slight or early renal dysfunction, and the accuracy of those markers in animal experiments is controversial [29]. Furthermore, we injected a large amount of BSA, which was metabolized to urea nitrogen, to introduce FAON. The assessment of renal function by BUN is, therefore, thought to be unsuitable in experiments with FAON model mice.

Interestingly, the amount of urine protein caused by FAON was dramatically reduced by PEM. The serum levels of total protein and albumin did not differ between the FAON and PEM + FAON groups (Appendix A), and no pathological abnormalities were observed in the glomeruli of the groups (Appendix A). These results imply that the amount of protein that reached the proximal tubular lumen was similar in both groups and that the reabsorption ability in PTECs accounted for the difference in urine protein amount. The reabsorption mechanism consumes a considerable amount of energy [3], mostly supplied by FAM [30]. In the impaired PTECs of the FAON group, this reabsorption mechanism might have collapsed, and large amounts of non-reabsorbed protein were excreted in urine. In contrast, PEM presumably improved FAM and the reabsorption mechanism of PTECs in the PEM + FAON group and reduced urine protein.

The renal expressions of mRNA and proteins related to FAM and renal ATP content were decreased in the FAON group despite the elevation of renal FFA content, an energy substrate. This fact suggests that the LTx of excessive FFA caused defective FAM and PTEC energy homeostasis in the FAON group. Considering that FAM and energy homeostasis are crucial for PTECs [3] and that renal FAM and ATP content were improved along with a decrease in renal FFA content in the PEM + FAON group, the renoprotective effect of PEM might be mediated by eliminating excessive FFA and improving FAM and energy homeostasis in PTECs.

Lastly, since OS markers in the kidney were reduced in the PEM + FAON group, PEM might have improved tubular injury through antioxidative effects. Considering that FFA causes OS [5,31], PEM could have indirectly reduced OS by the elimination of renal FFA-induced LTx. Several studies have reported a direct antioxidative effect of PPARα activation as well [18,19], and Maki et al. witnessed that PEM decreased OS by inhibiting the DAG-PKC pathway in diabetic nephropathy model mice [32]. Similarly, in the present study, PEM maintained the renal expressions of mRNA and proteins of SOD2, an antioxidant enzyme, against FAON. Thus, the agent could protect the kidneys through both direct and indirect antioxidative routes in FAON model mice.

There are several limitations to this study. First, we employed a pretreatment protocol, and so it remains uncertain whether PEM exhibits renoprotective effects even after the establishment of tubular injury by FAON. Considering that the main mechanism underlying the renoprotective effect of PEM is the elimination of toxic factors, such as FFA, the administration of PEM after the establishment of tubular injury may have produced weaker renoprotection. However, considering the clinical use of PEM and the importance of treatment from an early stage of kidney injury [33], the renoprotective results of PEM in this pretreatment protocol study are important. Second, the duration of PEM administration was only 20 days; additional examination of the long-term effects of PEM is needed. Furthermore, the duration of FFA-BSA injection might have been short at 14 days, considering that serum Cr was not elevated by FAON despite severe pathological kidney damage. Since tubular injury is strongly associated with long-term renal prognosis [26], the current results of the tubular protective effects of PEM against LTx derived from proteinuria implies a long-term benefit of PEM administration for CKD. To assess the extended benefits of PEM, further studies with longer PEM administration and FFA-BSA injection protocols are necessary. Third, the precise mechanism of the renoprotective effect of PEM remains unclear. In our FAON mouse experiments, we observed that maintaining renal FAM and decreasing OS by PEM could attenuate kidney injury, although the mechanism of how PEM improved renal FAM and lowered OS was not fully analyzed. Indeed, PEM may have other effects that contribute to renal protection. Additional in vitro studies and trials with other kidney injury protocols are being planned.

## 4. Materials and Methods

### 4.1. Animals and Experimental Design

WT C57BL/6 J male mice purchased from CLEA Japan, Inc. (Tokyo, Japan) were used in this study. The mice were maintained in a specific pathogen-free facility, housed in a light- and temperature-controlled environment (12-hour light/dark cycle at 25 °C), and provided tap water and standard chow ad libitum. For experiment one, eight-week-old male mice (body weight: 21–25 g) were divided into two groups, a vehicle (0.1% methylcellulose; MC)-treated group (Veh, *n* = 8) and a PEM (0.5 mg/kg body weight/day)-treated group (PEM, *n* = 7). PEM was kindly provided by Kowa (Nagoya, Japan). PEM was dissolved in 0.1% MC, and 0.1 mL of the chemical was administered to mice via oral gavage daily at 9:00 a.m. After 14 days, the mice were deeply anesthetized with isoflurane and sacrificed. For experiment two, 13-week-old WT mice (body weight: 25–30 g) were administered a daily intraperitoneal bolus injection of FFA-BSA in sterile saline to induce FAON [17,18]. FFA-BSA was obtained from Sigma Chemical (St. Louis, MO, USA, cat #9048-46-8). The following mice groups were used: vehicle-treated mice with daily intraperitoneal bolus injection of sterile saline (CON, *n* = 8), vehicle-treated mice with FAON (FAON, *n* = 10), and PEM (0.5 mg/kg body weight/day)-treated mice with FAON (PEM + FAON, *n* = 7). The CON and FAON groups were compared to detect the pathological changes produced by FAON, and we compared the FAON and PEM + FAON groups to determine the pharmacological action of PEM against FAON. The vehicle or PEM was administered daily via oral gavage from day 1 to day 20, and sterile saline or FFA-BSA was injected intraperitoneally every day from day 7 to day 20. FFA-BSA was dissolved to 33% in sterile saline, and the injection dose of chemicals was started from 0.9 mL/day and increased gradually (0.9 mL/day from day 7 to day 10, 1.2 mL/day from day 11 to day 13, and 1.5 mL/day from day 14 to day 20). Urine samples for the total volume per day of each group were collected using metabolic cages (Tecniplast, Tokyo, Japan). All mice were deeply anesthetized and sacrificed on day 20. In both experiments, serum and kidney samples were harvested from each animal, and all samples were stored at −80 °C until analysis. 

### 4.2. mRNA Analyses

Total kidney RNA was extracted using an RNeasy Mini Kit (QIAGEN, Hilden, Germany). Five micrograms of RNA were reverse-transcribed using oligo (dT) primers and SuperScript III reverse transcriptase (Invitrogen, Carlsbad, CA, USA). Subsequently, relative mRNA amounts were quantified by real-time polymerase chain reaction (PCR) using a THUNDERBIRD Probe qPCR Mix (TOYOBO, Osaka, Japan) on a QuantStudio 3 real-time PCR system (Thermo Fisher Scientific, Waltham, MA, USA). The specific primers were designed as shown in Appendix A. The *Actb* gene was used as the internal control. The ΔΔCt method for PCR amplification was employed to calculate the relative expression of mRNA in each group compared with the Veh group in experiment one and with the CON group in experiment two.

### 4.3. Immunoblot Analyses

Nuclear and cytoplasmic fractions of whole kidney extracts were prepared from each mouse using Nuclear and Cytosolic Extraction Reagents (Thermo Scientific, Rockford, IL, USA). Protein concentrations were determined using a BCA protein assay kit (Thermo Fisher Scientific, Rockford, IL, USA). Nuclear (15 μg) and cytoplasmic (5 μg) proteins were separated by sodium dodecyl sulfate polyacrylamide gel electrophoresis and transferred to polyvinylidene fluoride membranes. After blocking, the membranes were incubated with primary antibodies, followed by horseradish peroxidase-AffiniPure goat anti-rabbit IgG (Jackson ImmunoResearch Laboratories, cat #111-035-144, Cambridge, UK). Immunoblotting was performed using primary antibodies against CPT2 [34], VLCAD [35,36,37], MCAD [2], TPα [38], ACOX [39], PH [39], PT [40], and catalase [41]. Primary antibodies against PPARα (cat #9000) and SOD1 (cat #11407) were purchased from Santa Cruz Biotechnology (Santa Cruz, CA, USA), those against 4-HNE (cat #210-767-R100) were from Alexis (Lausanne, Switzerland), and those against SOD2 (cat #06-984) were from Millipore (Billerica, MA, USA). β-actin (cat #8227), and TATA box-binding-proteins (cat #63766) were used as internal controls for cytoplasmic and nuclear proteins, respectively (Abcam, Cambridge, UK). The detection of protein signals was carried out with a ChemiDoc Touch Imaging System (Bio-Rad Laboratories, Hercules, CA, USA) with EZ west Lumi plus (ATTO, Tokyo, Japan). The positions of the protein bands were determined by Precision Plus Protein Standards (Bio-Rad Laboratories, Hercules, CA, USA). Band intensities were measured densitometrically using NIH ImageJ software (National Institutes of Health, Bethesda, MD, USA), and we calculated the relative expressions of protein in each group compared with the Veh group in experiment 1 and with the CON group in experiment 2.

### 4.4. Histopathological Analyses

Kidney tissues were fixed in 10% formaldehyde. Deparaffinized sections were stained with periodic acid-methenamine-silver and used for the evaluation of glomeruli and tubules using a CX41N-31 microscope (Olympus, Tokyo, Japan). The severity of tubular injury was scored by means of a grading system (0–3 points), as follows: 0 = no visible lesion or near normal; 1 = mild dilation in a few proximal tubules or slightly injured PTECs; 2 = obvious dilation of many tubules or many injured PTECs and debris in the lumen; and 3 = severe dilation and luminal cast formation. Ten images of non-overlapping fields from the cortical area were randomly selected, and their total scores were calculated. The scores were evaluated in a blinded manner by D.A. and Y.Y. Immunohistochemical staining with osteopontin was performed to evaluate tubular injury. Each paraffin-embedded section was deparaffinized in Hemo-De (FALMA, Tokyo, Japan) and rehydrated with decreasing concentrations of ethanol. Sections were heated in ethylenediaminetetraacetic acid buffer at pH 9.0 for 30 min in a microwave oven, and then cooled slowly. After blocking with 1% BSA in PBS, the sections were reacted overnight with anti-mouse osteopontin (Immuno-Biological Laboratories, cat #18621, Takasaki, Japan). Peroxidase-conjugated anti-mouse immunoglobulin (Histofine Simple Stain MAX PO, Nichirei Biosciences, Tokyo, Japan) was used as the secondary antibody, and then the sections were stained with a 3,3′-Diaminobenzidine solution for observation.

For electron microscopy analysis, 1 mm pieces of kidney tissues were left in 2.5% glutaraldehyde solution for 1 hour, and sections of 1 µm thickness from epoxy resin-embedded tissues were stained with 1% toluidine blue to detect PTEC sites by light microscopy. Sections of 60 nm thickness containing PTEC sites were stained with 1% uranyl acetate for 10 min, followed by incubation with lead citrate for 5 min at room temperature. The stained sections were examined using a JEM 1400 system (JEOL, Tokyo, Japan).

### 4.5. Other Methods

For the measurement of renal ATP content, a 30 mg kidney sample was homogenized in 600 μL of ultrapure water and centrifuged (10,000× *g* at 4 °C for 10 min). Then, the ATP concentration in the supernatant was measured using a Tissue ATP assay kit (Toyo B-Net, Tokyo, Japan) according to the manufacturer’s instructions. Blood tests were performed using a JCA-BM6070 clinical analyzer (JEOL, Tokyo, Japan). BUN, serum Cr, FFA, TG, total protein, albumin, and ALT levels were measured by standard methods. Serum HDL and low-density lipoprotein levels were evaluated by the direct homogeneous assay method. Urine tests were performed with a JCA-BM6050 clinical analyzer (JEOL, Tokyo, Japan). Urine total protein levels were determined by the pyrogallol Red assay method. Urine NAG and Cr concentrations were measured by the enzyme assay method. Renal FFA was extracted using the hexane/isopropanol method as previously described [42]. Briefly, a 20 mg kidney sample was homogenized in 100 μL of ice-cold phosphate-buffered saline. Lipids were extracted from the homogenate with 900 μL of *n*-hexane:isopropanol (3:2, *v/v*) and dried using an evaporator. The dried lipids were solved with a 0.1% solution of Triton X-100 in ultrapure water (100 μL), and then FFA content was measured by means of NEFA C-test kits (Wako Pure Chemical, Osaka, Japan).

### 4.6. Statistical Analyses

Values represent the mean ± standard error of mean. Differences between groups were analyzed using a Student’s *t*-test. In experiment two, only differences between the CON and FAON groups and the FAON and PEM + FAON groups were assessed. Values of *p* < 0.05 were considered statistically significant. Statistical analyses were performed using IBM SPSS software version 27.0 (IBM, Armonk, NY, USA).

## 5. Conclusions

PEM exerts renoprotective properties against kidney injury from FFA overload. The maintenance of renal FAM and reduction of OS are possible mechanisms for this effect. PEM represents a promising drug in patients with kidney disease.

## Figures and Tables

**Figure 1 metabolites-11-00372-f001:**
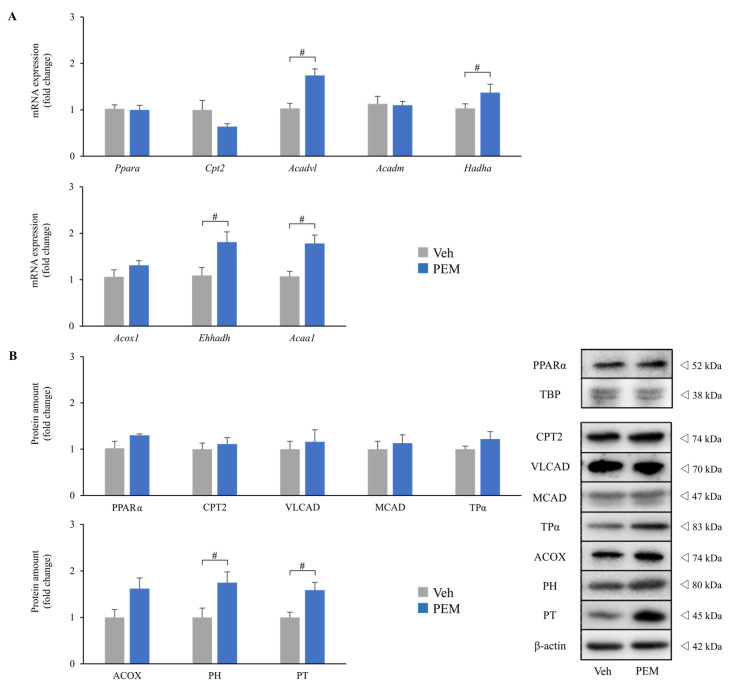
Effects of pemafibrate on renal fatty acid metabolism. (**A**) Expressions of renal mRNA related to fatty acid metabolism. (**B**) Renal protein levels related to fatty acid metabolism. Significant differences are indicated with # (*p* < 0.05). Abbreviations: *Ppara* (PPARα), peroxisome proliferator activated receptor alpha; TBP, TATA-binding protein; *Cpt2* (CPT2), carnitine palmitoyl-transferase 2; *Acadvl* (VLCAD), very long-chain acyl-CoA dehydrogenase; *Acadm* (MCAD), medium-chain acyl-CoA dehydrogenase; *Hadha* (TPα), mitochondrial trifunctional protein α subunits; *Acox1* (ACOX), acyl-CoA oxidase; *Ehhadh* (PH), L-peroxisomal bifunctional protein; *Acaa1* (PT), peroxisomal 3-ketoacyl-CoA thiolase; Veh, vehicle; PEM, pemafibrate.

**Figure 2 metabolites-11-00372-f002:**
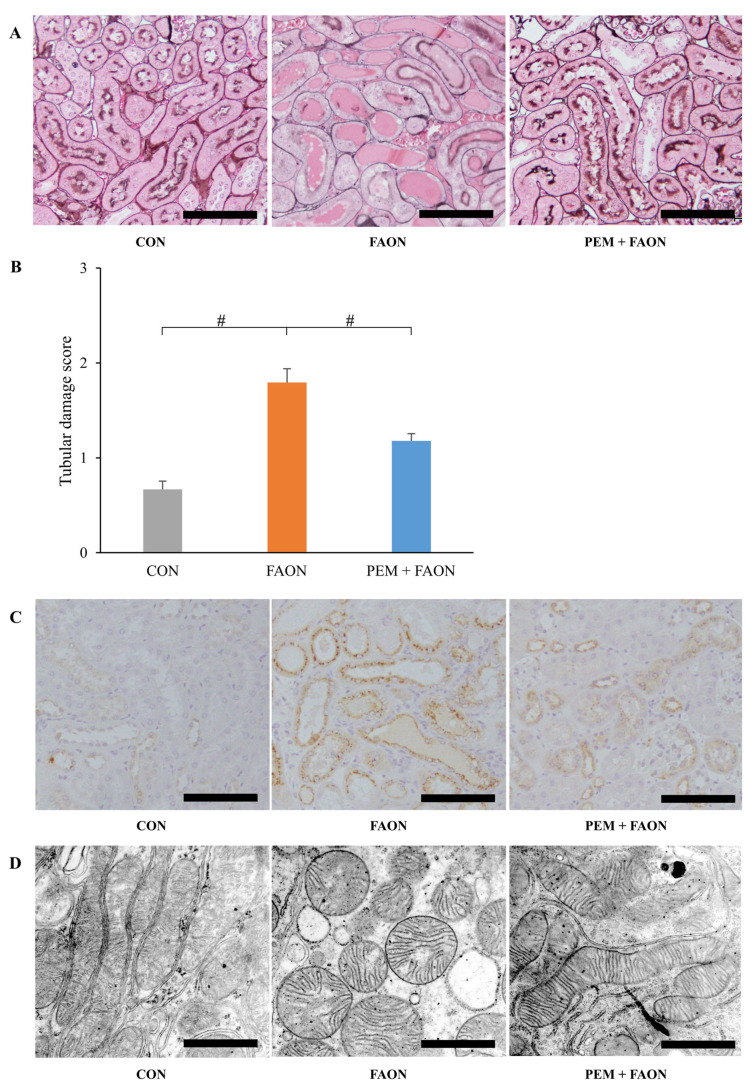
Pathological changes in renal tubular lesions by pemafibrate in experiment two. (**A**) Light microscopic analysis of tubular lesions. The sections were stained with periodic acid-methenamine-silver. Scale bar = 100 μm. (**B**) Damage scores of tubular lesions. Significant differences compared with the FAON group are indicated with # (*p* < 0.05). (**C**) Immunohistochemical staining for osteopontin. Scale bar = 100 μm. (**D**) Electron microscopic analysis of mitochondria in proximal tubular epithelial cells. Scale bar = 1 μm. Abbreviations: CON, control; FAON, fatty acid overload nephropathy; PEM, pemafibrate.

**Figure 3 metabolites-11-00372-f003:**
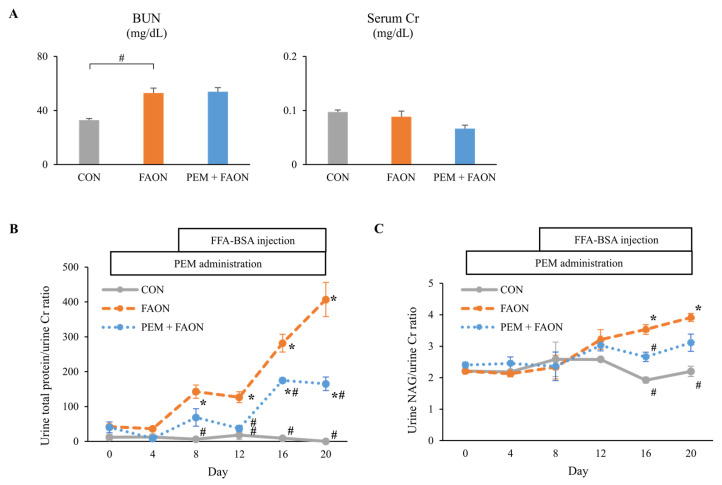
Changes in kidney injury markers by pemafibrate in experiment two. (**A**) Blood urea nitrogen and serum creatinine levels. (**B**) Urine total protein/urine creatinine ratio. (**C**) Urine N-acetyl-β-D-glucosaminidase/urine creatinine ratio. Significant differences compared with the FAON group are indicated with # (*p* < 0.05). In Figure 3B,C, significant differences compared with day 0 in each group are indicated with * (*p* < 0.05). Abbreviations: BUN, blood urea nitrogen; Cr, creatinine; NAG, N-acetyl-β-D-glucosaminidase; CON, control; FAON, fatty acid overload nephropathy; PEM, pemafibrate; FFA-BSA, free fatty acid-binding bovine serum albumin.

**Figure 4 metabolites-11-00372-f004:**
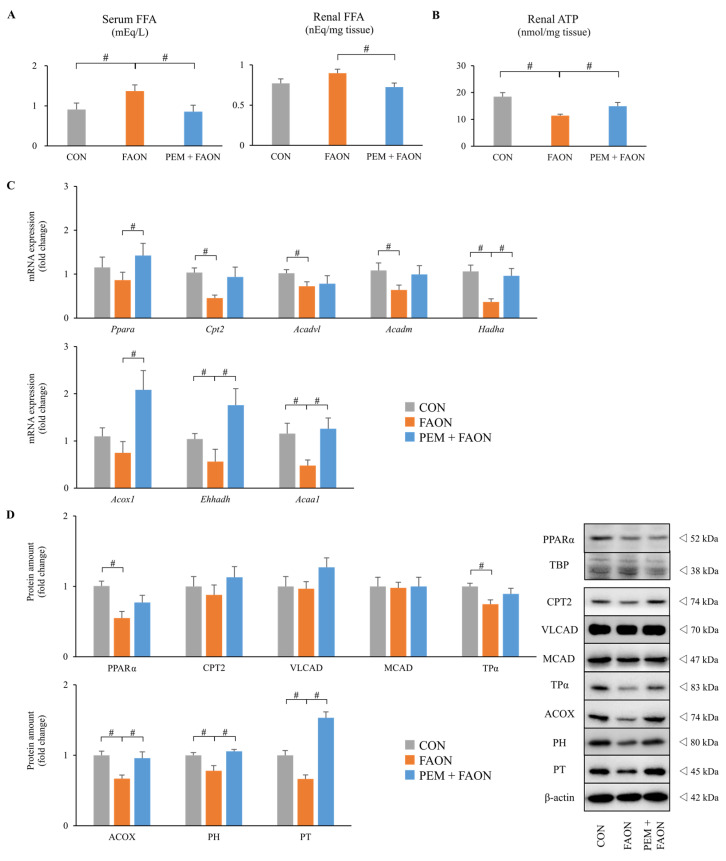
Changes in renal fatty acid metabolism by pemafibrate in experiment two. (**A**) Serum and renal free fatty acid levels. (**B**) Renal adenosine triphosphate content. (**C**) Renal expressions of mRNA related to fatty acid metabolism. (**D**) Renal protein levels related to fatty acid metabolism. Significant differences compared with the FAON group are indicated with # (*p* < 0.05). Abbreviations: FFA, free fatty acid; ATP, adenosine triphosphate; *Ppara* (PPARα), peroxisome proliferator activated receptor alpha; TBP, TATA-binding protein; *Cpt2* (CPT2), carnitine palmitoyl-transferase 2; *Acadvl* (VLCAD), very long-chain acyl-CoA dehydrogenase; *Acadm* (MCAD), medium-chain acyl-CoA dehydrogenase; *Hadha* (TPα), mitochondrial trifunctional protein α subunits; *Acox1* (ACOX), acyl-CoA oxidase; *Ehhadh* (PH), L-peroxisomal bifunctional protein; *Acaa1* (PT), peroxisomal 3-ketoacyl-CoA thiolase; CON, control; FAON, fatty acid overload nephropathy; PEM, pemafibrate.

**Figure 5 metabolites-11-00372-f005:**
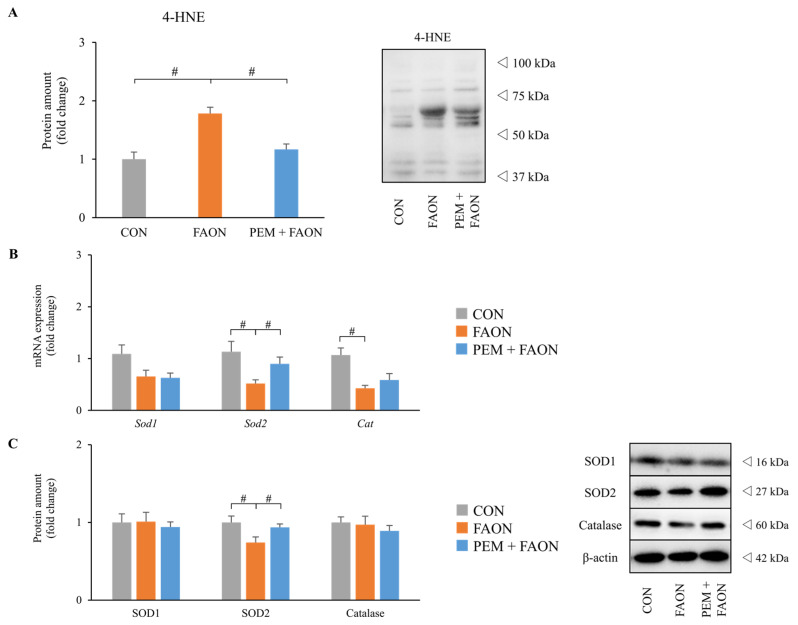
Changes in oxidative stress markers in the kidney by pemafibrate in experiment two. (**A**) Protein levels of 4-hydroxynonenal, an oxidative stress marker, in the kidney. (**B**) Renal expressions of mRNA related to antioxidant agents. (**C**) Renal expressions of proteins related to antioxidant agents. Significant differences compared with the FAON group are indicated with # (*p* < 0.05). Abbreviations: 4-HNE, 4-hydroxynonenal; *Sod1* (SOD1), superoxide dismutase 1; *Sod2* (SOD2), superoxide dismutase 2; *Cat* (Catalase); CON, control; FAON, fatty acid overload nephropathy; PEM, pemafibrate.

## Data Availability

Data are contained within the article or supplementary materials.

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
