# Peer review of "Pemafibrate Protects against Fatty Acid-Induced Nephropathy by Maintaining Renal Fatty Acid Metabolism"

_metabolites, 2021, doi:10.3390/metabo11060372_

Round 1

Reviewer 1 Report

  1. The quality of the figures needs to be improved.
  2. Pemafibrate (PEM) was first approved in Japan in 2017. It is very new. It is good to add the PEM group as control.
  3. Add more background of PEM in the section of the introduction.
  4. For the EM results, it is better to use higher power. 
  5. Many methods were not shown in the section of methods.
  6. I suggest the author should do more in vitro studies to confirm his conclusion and do more mechanism studies to make sure its functions.

Author Response

Response to Reviewer 1 Comments

Point 1: The quality of the figures needs to be improved.

Response 1: Thank you for pointing this out. We have replaced all figures with those of higher resolution.

Point 2: Pemafibrate (PEM) was first approved in Japan in 2017. It is very new. It is good to add the PEM group as control.

Response 2: We appreciate this suggestion. As the Reviewer stated, it would be ideal to add the PEM group as a control in experiment 2 and compare the 4 groups (CON, PEM, FAON, and FAON+PEM) to assess the precise effect of PEM on FAON. Indeed, the lack of comparisons with a PEM group in experiment 2 may reduce the robustness of our conclusion. However, it is impossible for us to newly add and analyze a PEM group in experiment 2 within the 3 weeks for revision.

We do think, however, that the results of experiment 1 partially cover this point. The effect of PEM on non-injured mice was assessed in experiment 1, and only serum HDL levels, serum TG levels, and gene expression related to fatty acid metabolism (FAM) were altered by PEM. We observed no obvious differences for renal histological appearance, renal ATP levels, or other serum markers. Based on those results, the PEM group in experiment 2 would presumably not exhibit any differences from the CON group apart from some FAM-related markers, which were already revealed as a key mechanism in the renoprotective effect of PEM by comparisons between the FAON and PEM + FAON groups. Taken together, we considered that the effects of PEM were revealed in experiment 1 and that the effects of FAON and of PEM against FAON were addressed in experiment 2. Although not precisely what was advised by the Reviewer, the results of experiment 1 may sufficiently represent the PEM group in experiment 2, especially in consideration of the revision timeframe.

We have added additional data of experiment 1, including histological kidney appearance, renal ATP levels, and renal mRNA expression related to anti-oxidative stress, as supplementary Figure 1. Descriptions on the above issues were inserted in lines 99 and 297.

Point 3: Add more background of PEM in the section of the introduction.

Response 3: We appreciate this advice and have added more information on PEM in the Introduction in line 51.

Point 4: For the EM results, it is better to use higher power.

Response 4: Thank you for pointing this out. We have replaced the EM figures with those of higher power (Figure 2D).

Point 5: Many methods were not shown in the section of methods.

Response 5: Thank you for raising this point. We have inserted additional descriptions on our methods for light microscopic analysis, immunohistochemical staining, specimen preparation for electron microscopy, blood tests, urine tests, and lipid extraction in lines 345, 352, 361, 372, 375, and 378, respectively.

Point 6: I suggest the author should do more in vitro studies to confirm his conclusion and do more mechanism studies to make sure its functions.

Response 6: We deeply appreciate these insights. As indicated by the Reviewer, further in vitro studies would confirm our conclusion and clarify the mechanism of the renoprotective effect of PEM. However, we are not able to conduct such studies within the 3 weeks for revision. We have added a description on the necessity of mechanism studies, including in vitro assays, as a limitation of this study in line 272.  

Reviewer 2 Report

In the paper novel  selective PPARα modulator - Pemafibrate - was investigater in preclinical, experimental setting.  Pemafibrate exerts renoprotective properties against kidney injury induced by FFA overload in mice.

In general style, methodology and quality of experiment is high. Results are displayed in selfexplanatory manner. Histopathology speciments confirm described effects of Pemafibrate.

Figures resolution seems on border of acceptance and I suggest to submit hihger resolution images before paper production.

Congratulations to authors

Author Response

Response to Reviewer 2 Comments

Point 1: In general style, methodology and quality of experiment is high. Results are displayed in self-explanatory manner. Histopathology speciments confirm described effects of Pemafibrate.

Response 1: Thank you for your favorable estimation of our paper.

Point 2: Figures resolution seems on border of acceptance and I suggest to submit higher resolution images before paper production.

Response 2: We appreciate this comment and have replaced all of the figures with ones of higher resolution.

Reviewer 3 Report

The manuscript describes an elegant study demonstrating the protective effect of PEM on fatty acid-induced renal dysfunction.

REnal function has been addressed after high fatty acid diet in the presence of PEM and vehicle, by using immunoistochemistry studies, functional analysis ( plasmatic cretinine and urea plus urine protein excretion) and mRNA and protein levels of selected enzymes. 

I have only few concerns:

PEM seems protective, as suggested by data shown.

However, in oder to confirm the protective effect on long term, I would suggest to show longer time points of drug treatment, when mice GFR declines in controls. This data would support the protective role of PEM in chronic conditions, which should be the main message for readers.

Author Response

Response to Reviewer 3 Comments

Point 1: PEM seems protective, as suggested by data shown. However, in order to confirm the protective effect on long term, I would suggest to show longer time points of drug treatment, when mice GFR declines in controls. This data would support the protective role of PEM in chronic conditions, which should be the main message for readers.

Response 1: Thank you very much for these insights. As the Reviewer states, the assessment of longer time points is needed to confirm the renoprotective effect of PEM. In this regard, the 20-day PEM administration period was insufficient. The duration of FFA-BSA injections (14 days) may also have been short considering that serum Cr levels were not elevated by FAON despite severe pathological damage in kidneys. However, we regret that it is not possible for us to newly perform such additional experiments within the 3 weeks for revision.

In our study, however, renal interstitial damage, which has been shown as a strong predictive marker of renal prognosis, was clearly evoked by FAON and attenuated by PEM. These results support the possibility that PEM protects from interstitial damage, leading to a long-term renoprotective effect.

We agree that further study with long-term experiments is absolutely needed to establish the renoprotective effects of PEM. We have added a description on this as a study limitation in line 264. 

Round 2

Reviewer 3 Report

Major comments have been addressed; limitations are stressed in the revised manuscript